# The Relationship between Sleep Duration and Metabolic Syndrome Severity Scores in Emerging Adults

**DOI:** 10.3390/nu15041046

**Published:** 2023-02-20

**Authors:** Bilal A. Chaudhry, Michael S. Brian, Jesse Stabile Morrell

**Affiliations:** 1Department of Kinesiology, University of New Hampshire, Durham, NH 03824, USA; 2Department of Agriculture, Nutrition, and Food Systems, University of New Hampshire, Durham, NH 03824, USA

**Keywords:** sleep, sleep duration, metabolic syndrome, metabolic syndrome severity score, emerging adulthood, diabetes, hypertension, obesity, dyslipidemia

## Abstract

Background: Research suggests sleep duration can influence metabolic systems including glucose homeostasis, blood pressure, hormone regulation, nervous system activity, and total energy expenditure (TEE), all of which are related to cardiometabolic disease risk, even in young adults. The purpose of this study was to examine the relationship between sleep duration and metabolic syndrome severity scores (MSSS) in a sample of emerging adults (18–24 y/o). Methods: Data were collected between 2012 and 2021 from the College Health and Nutrition Assessment Survey, an ongoing, cross-sectional study conducted at a midsized northeastern university. Anthropometric, biochemical, and clinical measures were obtained following an overnight fast and used to assess the prevalence of metabolic syndrome (MetS). MetS severity scores (MSSS) were calculated using race- and sex-specific formulas. Sleep duration was calculated from the difference in self-reported bedtime and wake time acquired through an online survey. ANCOVA was used to examine the relationship between sleep duration and MetS severity score while adjusting for covariates (age, sex, BMI, physical activity level, smoking status, alcohol consumption, and academic major). Results: In the final sample (*n* = 3816), MetS (≥3 criteria) was present in 3.3% of students, while 15.4% of students presented with ≥2 MetS criteria. Mean MSSS was −0.65 ± 0.56, and the reported sleep duration was 8.2 ± 1.3 h/day. MSSS was higher among low sleepers (<7 h/day) and long sleepers (>9 h/day) compared to the reference sleepers (7–8 h/day) (−0.61 ± 0.02 and −0.63 ± 0.01 vs. −0.7 ± 0.02, respectively, *p* < 0.01). Conclusions: Our findings suggest short (<7 h/day) and long (>9 h/day) sleep durations raise the risk of MetS in a sample of emerging adults. Further research is needed to elucidate the impact of improving sleep habits on future disease risk.

## 1. Introduction

Metabolic syndrome (MetS) is a clustering of metabolic risk factors that leads to a ~2-fold increase in cardiovascular disease and a ~3.5-fold increase in diabetes [1,2]. In 2016, it was reported that approximately 36.9% of adults in the U.S. met MetS criteria, reflecting a 46% increase from 25.3% in 1999 [3,4]. More specifically, prevalence of metabolic syndrome in individuals 20–39 years old increased from 16.2% in 2011 to 21.3% in 2016 [3]. This significant increase in the prevalence of MetS in younger adults raises concerns about the effects of modifiable lifestyle factors on metabolic health. MetS is defined by the total number (≥3) of risk factors (i.e., elevated fasting blood glucose, hypertension, elevated triglycerides, low HDL-C, and central obesity), which provides a binary classification for patients. However, the binary diagnosis does not represent the severity of MetS [5,6]. A novel technique developed by Lee and colleagues (2016) attempts to calculate the severity of MetS while incorporating gender and racial differences [7]. This metabolic syndrome severity score (MSSS) offers advantages in clinical populations to track the progression of the condition as well as the efficacy of lifestyle interventions. Previous work using data from the 2013/2014 National Health and Nutrition Examination Survey (NHANES) has linked sleep duration with MSSS [8]. However, as young adults, specifically college students, are frequently absent from national surveillance data due to study protocols, there remains a gap in understanding the relationship between sleep and metabolic health in the young adult population.

Emerging adulthood is a stage of life filled with many transitions and “firsts”, often associated with moving away from parents/guardians and the childhood home [9,10,11]. Whether living independently or cohabiting with peers, emerging adulthood is often the first time many young adults assume control of their daily schedules and lifestyle choices. This newfound autonomy can lead to significant lifestyle changes, which may be more likely to be maintained into adulthood [12]. Further, lifestyle habits have been closely linked with metabolic health [13]. Insufficient sleep is reported among young adults, especially students, who sacrifice adequate sleep to study more and/or socialize [14]. Poor sleep can lead to habits that may have greater implications for overall health and well-being.

Adequate sleep is increasingly recognized as an important, modifiable lifestyle factor, given its role in metabolic and neurological health [15]. Low sleep (4 h/night) has been found to decrease vagal tone and increase sympathetic activity, which can ultimately lead to insulin resistance, hypertension, and obesity [15]. Research in middle-aged adults has demonstrated a link between sleep patterns and MetS [16,17,18,19,20,21,22,23,24]. Smiley and colleagues (2019) found a U-shaped relationship between sleep duration and prevalence of metabolic syndrome and MetS severity in adults 18–80 years of age [8]. However, it remains unknown if sleep patterns during emerging adulthood (18–24 y/o) are linked to the prevalence and severity of MetS.

Self-reported sleep duration on school nights may decline during adolescence. Carskadon et al. (2011) observed an average of 8.4 h a night among sixth graders and only 6.9 h a night among twelfth graders [25]. Furthermore, research indicates that college students, in particular, suffer from poor sleep due to a variety of factors, including delayed bedtimes and rise times on the weekends, as well as the use of medications that may alter the sleep/wake cycle [26]. Emerging adults are introduced to many new stressors in college life, many of which are associated with newfound independence [27].

The purpose of this study was to examine the association between sleep duration and the prevalence of MetS in emerging adults (18–24 y/o). Additionally, we examined the relationship between sleep duration and MetS severity scores. We hypothesized that short and long sleep duration would be associated with greater MetS severity scores. Further, we hypothesize the relationship will be amplified in young adults with MetS (i.e., ≥3 risk factors).

## 2. Materials and Methods

Data were collected between 2012 and 2021 from a convenience sample of student participants between the ages of 18 and 24 years, enrolled in a general education, introductory nutrition course. As detailed in previous work, participants were recruited during the fall and spring semesters from the large, multi-section lecture/laboratory course as part of an ongoing, cross-sectional College Health and Nutrition Assessment Survey at the University of New Hampshire [28,29,30,31]. All methods were conducted in accordance with the Declaration of Helsinki and approved by the Institutional Review Board (UNH IRB #5524). Participants were provided an overview of the study design and objectives on the first day of class; written, informed consent was obtained before any data collection. All research staff were trained in collection methods, as well as research ethics and data management, prior to data collection. Participants were provided a brief explanation prior to each assessment and given the opportunity to ask questions as needed. Participants were not provided with any incentives.

### 2.1. Anthropometric Measures

Participants self-scheduled an in-person visit for the assessment of anthropometric, biochemical, and clinical measures during the 6th or 7th week of the 15-week semester. Participants were instructed to arrive at the testing site in the morning following a 10–12 h overnight fast. Participants were instructed to avoid vigorous activities for at least 4 h and alcohol for at least 48 h, as well as void immediately before their scheduled appointment. Measures were conducted without shoes and with participants wearing t-shirts and shorts in semi-private, partitioned spaces. Height (cm) was measured during inhalation via a wall-mounted stadiometer (Accustat Genetech, San Francisco, CA, USA; Seca 264, Chino, CA, USA) following standard protocols, and body mass (kg) was measured via digital scale (#2000A, Life Measurement Inc. Concord, CA, USA; Tanita BWB-800, WB-800S, Arlington Heights, IL, USA). Waist circumference (cm) was measured during exhalation via Gulick tape measure at the iliac crest following the protocol described by the National Health and Nutrition Examination Survey (NHANES) [32]. Height, body mass, and waist circumference measures were each collected in duplicate, and averaged. All equipment was calibrated each morning before data collection.

### 2.2. Biochemical and Clinical Measures

Fasted blood samples were acquired via fingerstick and analyzed by the Alere Cholestech LDX Analyzer (Cholestech LDX Analyzer, Abbott; Chicago, IL, USA) for blood glucose, high-density lipoprotein cholesterol (HDL-C), and triglyceride concentrations. After measurement of mid-arm circumference, systolic blood pressure (SBP) and diastolic blood pressure (DBP) were measured twice in the right arm via an automated, appropriately sized cuff (HEM-711DLX; Omron, Bannockburn, IL, USA) after the participant was seated quietly and comfortably for a minimum of 5 min with feet placed flat on the floor. The average of the two blood pressure measurements was used for data analysis.

### 2.3. Metabolic Syndrome Measures

The prevalence of MetS was assessed using criteria for clinical diagnosis according to the standards established by the American Heart Association/National Heart, Lung, and Blood Institute to ensure accuracy and consistency with the previous literature [8,33,34,35]. Participants who met three or more of the following criteria were identified as having MetS: (1) elevated triglycerides, >150 mg/dL; (2) low HDL-C, <40 mg/dL for men and <50 mg/dL for women; (3) elevated blood pressure, >130/85 mmHg; (4) elevated fasting blood glucose, >100 mg/dL; and/or (5) elevated waist circumference, >102 cm for men and >88 cm for women [34].

### 2.4. Metabolic Syndrome Severity Score

For this study, we utilized the MSSS to assess the severity of MetS. The MSSS factors the traditional MetS definition criteria (waist circumference, SBP, fasting glucose, triglycerides, and HDL-C) while also accounting for sex, race, and ethnicity to calculate a z-score [7]. As race-specific formulas have not been developed for those who identify as American Indian/Alaskan Native or Asian/Pacific Islanders, we utilized the same formula used for non-Hispanic white males and females, respectively, for these individuals, as recommended by one of the study’s co-authors (personal communication). The calculations are provided in Table 1. The equations provide a standardized z-score, with the mean set to zero and a range from negative infinity to positive infinity. A greater severity score is associated with higher levels of MetS markers.

### 2.5. Assessment of Sleep Duration and Physical Activity Level

Sleep duration, physical activity, and other demographic information were self-reported as part of a longer (90+ item) online health behavior questionnaire (Qualtrics). Four questions assessed sleep duration and quality; self-reported bedtime and wake time were derived from “what time of day do you usually go to bed on most days of the week” and “what time of day do you usually get up on most days of the week”, and used to calculate sleep duration and midpoint. Participants were categorized into four defined sleep duration groups: Low Sleep (<7 h), Reference group #1 (7–8 h), Reference group #2 (8–9 h), and Long Sleep (>9 h). Cut-offs were selected based on previous literature that set 7–8 h/night or 8–9 h/night as reference groups [20,35,36,37]. Sleep quality was measured via survey item; participants were asked: “How often do you think you get enough sleep?”, and chose from: always, usually, sometimes, rarely, never, or I prefer not to answer. Physical activity level was measured via multiple items, including participants’ self-report frequency (days/week) and duration (minutes/day) of moderate and vigorous exercise participation in the past 7 days [38]. Participants were prompted to “think about the activities you do at work, as part of your house and yard work, to get from place to place, and in your spare time for recreation, exercise, or sport”. For vigorous activity, participants were prompted to report on activities such as, “heavy lifting, digging, aerobics, or fast bicycling”. For moderate activity, participants were prompted to “think about activities that take moderate physical effort and make you breathe somewhat harder than normal”. These responses were compared to the American College of Sports Medicine (ACSM) guidelines of physical activity: 150 min of moderate-intensity exercise, 75 min of vigorous-intensity exercise, or a mixture of both [39,40]. From these responses, a binary value was created to indicate whether or not participants met ACSM guidelines. Housing status was measured via one survey item where participants were asked: “Where do you currently live” and chose from: college-affiliated dorm (no kitchen in room), college-affiliated apartment (with kitchen), apartment (with kitchen), at home (permanent residence), I choose not to answer, or other.

### 2.6. Data Management and Statistical Analyses

All collected data were screened for errors and/or corrected when possible. Participants who reported being outside of the age parameters, being pregnant, use of medication that influences sleep–wake cycle (e.g., beta-blockers), or had missing anthropometric, biochemical, clinical, or fitness measurements were excluded from analysis. Missing data were most commonly associated with absence/failure to attend the in-person assessment. Further, individuals who did not answer sleep items or racial identity questions (n = 102) were excluded from the analysis.

Data are reported untransformed and presented as frequencies or means ± standard deviation. Prior to analyses, data were examined for normality via Levene’s Test (*p* = 0.129). One-way ANOVA, *t*-test, and Tukey’s test were used to evaluate significant differences between demographic characteristics and sleep duration subgroups. Analysis of covariance (ANCOVA) was used to compare differences in MSSS between sleep duration groups with age, sex, semester (fall/spring), BMI, binary physical activity value, smoking status, and academic major serving as covariates. Additionally, a sub-analysis was performed to determine the influence of sleep duration on MSSS in participants with MetS (≥3 criteria). Significance was set as *p* < 0.01 due to the large sample size; all statistical analyses were performed by SPSS v. 27.

## 3. Results

Our final sample included 3816 participants (31.2% male/68.8% female), with a mean age of 18.8 ± 1.1 years. The sample included a majority of participants with a normal BMI (mean 23.5 ± 3.7 kg/m^2^), nonsmokers (95.1%), non-Hispanic white (92.2%), non-allied health/nutrition majors (75.9%), met ACSM physical activity guidelines (88.1%), living in a college-affiliated dorm with no kitchen in the room (72.5%), and first-year students (55.6%). Participant demographics among the four sleep duration groups are presented in Table 2. About half of the sample (51.6%) had at least 1 criterion of MetS; 15.5% had at least 2 criteria of MetS, and 3.3% (n = 124) of the young adult sample met the criteria for MetS. Low HDL-C was the most commonly met MetS criteria (49%), while elevated glucose was the least commonly met MetS criteria (8%). The frequency of the individual MetS criteria is presented in Table 3.

The mean sleep duration per night was 8.2 ± 1.3 h for the entire sample; while the mean sleep duration was 6.1 ± 0.7, 7.3 ± 0.3, 8.3 ± 0.3, 9.5 ± 0.7 h for Low Sleep, Reference #1, Reference #2, Long Sleep, respectively. Mean bedtime for the sample was 11:49 p.m. ± 1:04, the average rise time was 8:01 a.m. ± 1:06, and average sleep midpoint was 3:55 a.m. ± 0:53. In regard to sleep quality, 12.1% of the whole sample reported either “usually” or “always” getting enough sleep, while 34.8% reported “sometimes”, 45.1% reported “rarely”, and 8% reported “never” getting enough sleep. We observed longer average sleep durations in women vs. men and non-health majors vs. allied health/nutrition majors; we observed no difference in smoking status and freshman status, respectively (women: 8.3 ± 1.2 vs. men: 8.1 ± 1.3 h, *p* < 0.01; non-health majors: 8.3 ± 1.3 h vs. allied health/nutrition majors: 8.0 ± 1.2, *p* < 0.01; smokers: 8.1 ± 1.4 vs. non-smokers: 8.2 ± 1.2 h, *p* = 0.169; first year students: 8.2 ± 1.3 vs. upper class students 8.2 ± 1.2 h, *p* = 0.644; respectively).

For the entire sample, mean MSSS and sleep duration were −0.65 ± 0.56 and 8.2 ± 1.3 h per day, respectively. We observed equal variances throughout all sleep groups (*p* = 0.129). We observed an increase in MSSS between low sleepers (<7 h) and long sleepers (>9 h) compared to the reference #1 sleepers (7–8 h/day) (*p* < 0.01), but not reference #2 sleepers (*p* > 0.01). Gender, BMI, and the binary physical activity value were shown to significantly influence MSSS (for all *p* < 0.01; partial eta squared $\eta_{p}^{2}$ = 0.063; $\eta_{p}^{2}$ = 0.286, $\eta_{p}^{2}$ = 0.004; respectively). Those categorized in the Low Sleep group (<7 h) had higher fasting blood glucose and SBP compared to the Reference #1 group (Low Sleep: 87.1 ± 10.8 vs. Reference #1: 85.1 ± 10.1 mg/dL, *p* = 0.007; Low Sleep: 118 ± 12 vs. Reference #1:115 ± 12 mmHg, *p* = 0.002; respectively). However, we saw no significant difference in waist circumference, fasted HDL-C, and DBP between all groups (Low Sleep: 81.4 ± 9.9 vs. Reference #1: 80.6 ± 10.0 cm, *p* = 0.449; Low Sleep: 55.4 ± 15.9 vs. Reference #1: 54.7 ± 14.0 mg/dL, *p* = 0.862; Low Sleep: 73 ± 9 vs. Reference #1: 71 ± 9 mmHg, *p* = 0.034; respectively). Further, those categorized in the Long Sleep group (>9 h) had higher triglyceride levels vs. Reference #1 group (Long Sleep: 103 ± 44 vs. Reference #1: 96 ± 44 mg/dL, *p* < 0.001).

In the sub-analysis of participants who met ≥ 3 of the MetS criteria (n = 124; 3.3% of total sample), we observed no significant relationship between sleep duration and MSSS (r^2^ = 0.011, *p* = 0.92). Additionally, we observed no significant difference in individual MetS criteria in this group. In Table 4, we report positive MSSS for all categories of sleep groups.

## 4. Discussion

Our analysis of sleep duration and MSSS confirms and expands on the previous studies that indicate that both sleep loss and sleep extension are associated with worsening MetS severity. Importantly, our findings found that low (<7 h/day) and long (>9 h/day) sleep durations lead to greater MSSS scores in emerging adults, even after adjusting for confounding factors. To the best of our knowledge, this is the first study to examine the relationship between sleep duration and metabolic disease risk in an emerging adult population. These findings bring attention to the effect of sleep on metabolic biomarkers during a formative stage of life. As the behaviors that individuals acquire in this stage of life are often retained throughout adulthood, the data raise attention to the relationship between poor lifestyle habits and physical health and wellbeing [41].

Our study found a similar U-shaped curve relationship between sleep duration and MSSS as previous studies have reported in a variety of age groups [8,36,37,42,43]. Inadequate sleep hygiene has been associated with increased sympathetic nervous system activation, which is speculated to contribute to the development of metabolic dysfunction through multiple pathogenic pathways, including decreased cerebral glucose concentration, and elevated circulating glucose, ultimately leading to insulin resistance and obesity [19,36,42,44,45,46,47,48,49]. Similarly, sleep extension is typically associated with elevated cortisol levels, which are linked to hypertension and impaired pancreatic beta cell function, leading to insulin resistance and obesity [36].

Obesity has been strongly linked to poor sleep hygiene. A study analyzing NHANES data from 1982–1984, 1987, and 1992 found a negative relationship between sleep duration and BMI [44]. A more recent study confirmed that low (≤6 h/day) and long sleepers (≥9 h/day) were associated with greater increases in visceral adipose tissue after a 6-year follow-up, likely the result of previously mentioned increased energy balance and reduced physical activity associated with both low and long sleep durations [15,42]. These changes in BMI and waist circumference are associated with hormonal changes associated with low and long sleep, which can lead to a positive energy balance. Specifically, data from the Wisconsin Sleep Cohort Study found habitual sleep duration below 7.7 h/night was associated with decreased leptin, increased ghrelin, and a greater BMI [43]. Sympathetic activity inhibits leptin release and decreases vagal activity, which can lead to increased levels of ghrelin [50]. Previous literature suggests leptin deficiency is correlated with chronic sleep restriction, while elevated ghrelin is an acute response to sleep loss [50]. Nonetheless, the changes in both of these hormones can lead individuals to enter a positive energy balance [15]. Further, Taheri et al. (2004) found that long sleep was also associated with a greater BMI, likely the result of reduced energy expenditure due to increased time in bed as well as decreased rates of self-reported exercise per week [43]. Surprisingly, our study did not observe significant differences in BMI or waist circumference between sleep duration groups. This may be explained by our younger and fairly active participants, whose overall weight status is lower than that of older adult populations.

Inadequate sleep has been shown to alter glucose metabolism and increase the risk of developing type 2 diabetes [36,48,51]. Our analysis found that low sleepers (<7 h/day) had significantly higher mean fasted glucose when compared to Reference Group #1 (7–8 h/day). Others have linked sleep deprivation to diminished cerebral glucose utilization with associated increased insulin resistance [36]. In addition, the odds of meeting the glucose criteria for type 2 diabetes have been found to be at least 1.7× greater in very low (<6 h/day) and long (>8 h/day) sleep groups [19]. Collectively, these findings suggest sleep duration can influence an individual’s risk for developing type 2 diabetes and MetS. Increased glucose utilization during sleep has also been associated with REM sleep and waking due to the “dawn phenomenon”, when compared to non-REM sleep. Those with restricted sleep will likely experience fewer REM cycles, thus leading to glucose dysregulation [43]. Contrary to the findings of Gangwisch et al. (2007), we did not observe differences in fasting glucose between the reference and the long sleep groups [36]. This again may be due to our younger, active population, who are less likely to experience these adverse metabolic responses associated with aging.

Poor sleep hygiene is linked to elevated blood pressure and dyslipidemia. Studies have found that sleep restriction increases 24-h blood pressure, heart rate, sympathetic nervous system activity, and sodium retention [45,52]. Our study found that low sleep (<7 h) significantly increased mean SBP when compared to Reference Group #1 (7–8 h). Further, serum triglycerides have been shown to have a u-shaped relationship with sleep duration, while HDL cholesterol has an inverse u-shaped relationship [38]. Our study found that long sleep (>9 h) increased mean serum triglyceride levels when compared to Reference Group #1 (7–8 h), but no significant relationship between sleep duration and mean HDL-C levels were observed. Elevated triglycerides may be related to less healthy dietary intakes observed in long sleepers [53]. Overall, our findings support the hypothesis that inadequate sleep hygiene can have a cascading effect on all the factors of MetS (waist circumference, blood pressure, fasting blood glucose, triglyceride, and HDL-C levels), likely increasing an individual’s risk for developing MetS.

In our study, we also analyzed a subgroup that met MetS criteria. Here we observed positive MSSS values, which are consistent with clinical populations. However, there were no differences between sleep duration groups. This could be the result of the relatively small subsample size or negative risk factors associated with this young and fairly active population. Further research should explore this relationship in a less active cohort, where there may be a greater relationship.

Current recommendations for target lifestyle modifications for the treatment of MetS include dietary and activity interventions. Our findings suggest that sleep may be another lifestyle modification to include in these recommendations. To this end, future intervention studies that incorporate sleep duration are needed. Previous research suggests targeting sleep duration may reduce the severity of MetS. For example, in a cohort of healthy men, Spiegel et al. (1999) induced 6 days of sleep restriction with 4 h bedtimes followed by 7 days of sleep recovery [24]. After sleep restriction, subjects experienced reduced glucose tolerance, however, this was attenuated after the sleep recovery period. Further, in a cohort of twelve healthy non-obese individuals, 14 days of sleep restriction caused a significant increase in weight and subcutaneous and visceral abdominal fat, which returned to normal after 3 days of recovery sleep [54]. Zhu et al.’s 2019 meta-analysis found that sleep restriction led to alterations in the brain’s reward and cognitive control regions in response to eating, primarily through altering the regulation of leptin and ghrelin, and leading to increased food intake [55]. While interventions to improve sleep duration and quality have been varied and multifactorial, incorporating such strategies as the addition of dietary supplements (e.g., vitamin D, melatonin) and/or limiting alcohol or other stimulants (e.g., tobacco, caffeine), optimizing sleep may be an important target to improve metabolic and cardiovascular health, and future work is warranted [43,56,57,58,59,60].

A major strength of the current study is that it examined the relationship between an important lifestyle behavior and metabolic health in a young adult population during a pivotal point in their development. Given the varying sleep recommendations throughout the lifespan, our use of a distinct age range (18–24 y/o), as well as a large sample size, strengthens the utility of our findings [8,61,62]. Furthermore, our study utilized MSSS, which provides a weighted measure for addressing disease severity. The traditional method is a binary approach that assumes each component of MetS holds equal weight and does not account for the severity of the condition or how MetS manifests differently based on gender, race, and ethnicity. The MSSS accounts for these variables and allows us to track the progression of MetS over time [7]. This technique offers an advantage when analyzing the effects of different behaviors or interventions on the progression of MetS [8]. Another strength of this study was the large sample size (i.e., 3869 participants), reducing the margin of error compared to data in smaller studies.

Future research in this area should consider the limitations of our study. First, the cross-sectional data were obtained from a convenience sample, and, therefore, causality between variables cannot be established. However, this convenience design reduced participant burden and produced a high rate of participation among recruited individuals (>90%) who may not otherwise devote the time to the research activity. Second, the majority of our cohort consists of students living in college-affiliated dorms without a kitchen in the room, likely leading them to rely on dining halls for their dietary needs. This model may not adequately represent the effects on those in this age group that are living in or outside the college environment with more autonomy over their dietary options. Further, this study utilized our own survey items to acquire estimated bedtimes and wake times. Future research may consider the use of a validated survey (e.g., Pittsburgh Quality Sleep Index) to measure self-reported sleep duration [16,20]. Nonetheless, the wording of our questions did not significantly differ from that of the Pittsburgh Quality Sleep Index and is similar to other studies which utilized their own questionnaires [8,20,33,35,36,37,42,44,45]. We also did not account for chronotypes, which are defined as an individual’s circadian preference that describes their inclination for earlier or later timing sleep, and activities of daily living [63]. Sleep chronotypes have been shown to influence insulin sensitivity and glycemic control [64]. Lastly, we did not use direct measures of sleep duration, such as polysomnography, as used in other studies [23,43,65]. Subjective self-reported sleep durations have been shown to be over-estimated compared to objectively measured sleep duration measurements in older adults [66]. Therefore, objectively measured sleep duration and quality may yield a greater ability to assess the role sleep hygiene plays with the development of Metabolic Syndrome in emerging young adults.

Future studies may consider validating reported sleep behaviors via the use of polysomnography while also acquiring other variables of sleep quality, such as time spent in REM sleep, sleep continuity, HR, and breathing during sleep, as well as overall brain activity. Lastly, MSSS formulas have not been developed for those who identify as American Indian/Alaskan Native or Asian/Pacific Islanders. While only 3.6% of our population identified as one or more of these races/ethnicities, future research needs to adequately quantify the proper formulas based on risk so we can accurately calculate MSSS for these populations and elucidate the impact of improving sleep habits on future disease risk.

## 5. Conclusions

In conclusion, our findings suggest a relationship between sleep duration and elevated metabolic syndrome severity. More specifically, emerging adults sleeping less than 7 h or more than 9 h per night are at a disproportionate risk for developing cardiovascular disease and diabetes. This study highlights the need for effective assessments, education, and policies related to healthy lifestyle habits in young adults in order to attenuate the onset and progression of metabolic syndrome and the burden of future chronic disease. This work provides evidence for future research to incorporate sleep, along with diet and activity, to the lifestyle intervention strategies utilized in reducing the severity of metabolic dysfunction and to better understand the effects of sleep on metabolic health.

## Figures and Tables

**Table 1 nutrients-15-01046-t001:** Formulas utilized to calculate metabolic syndrome severity scores (MSSS) (z-score) for adult populations.

Non-Hispanic White Man
z-score = −5.4559 + 0.0125 × [WC] − 0.0251 × [HDL] + 0.0047 × [SBP] + 0.8244 × [log (TRG)] + 0.0106 × [GLU];
Non-Hispanic African-American Male
z-score = −6.3767 + 0.0232 × [WC] − 0.0175 × [HDL] + 0.0040 × [SBP] + 0.5400 × [log (TRG)] + 0.0203 × [GLU];
Hispanic Male
z-score = −5.5541 + 0.0135 × [WC] − 0.0278 × [HDL] + 0.0054 × [SBP] + 0.8340 × [log (TRG)] + 0.0105 × [GLU];
Non-Hispanic White Female
z-score = −7.2591 + 0.0254 × [WC] − 0.0120 × [HDL] + 0.0075 × [SBP] + 0.5800 × [log (TRG)] + 0.0203 × [GLU];
Non-Hispanic African American Female
z-score = −7.1913 + 0.0304 × [WC] − 0.0095 × [HDL] + 0.0054 × [SBP] + 0.4455 × [log (TRG)] + 0.0225 × [GLU];
Hispanic Female
z-score = −7.7641 + 0.0162 × [WC] − 0.0157 × [HDL] + 0.0084 × [SBP] s + 0.8872 × [log (TRG)] + 0.0206 × [GLU];

WC, Waist Circumference (cm); HDL, High-Density Lipoprotein (mg/dL); SBP, Systolic Blood Pressure (mmHg); TRG, Triglycerides (mg/dL); Glu, Glucose (mg/dL); cm, centimeters; mg/dL, milligrams per deciliter; mmHg, millimeters of mercury.

**Table 2 nutrients-15-01046-t002:** Participant demographics.

	Low Sleep (Less Than or Equal to 6.99 h)			Reference #1 (7–7.99 h)			Reference #2 (8–8.99 h)			Long Sleep (Greater Than or Equal to 9 h)		
	n = 453			n = 894			n = 1230			n = 1292		
	Mean		Standard Deviation	Mean		Standard Deviation	Mean		Standard Deviation	Mean		Standard Deviation
Age (years)	19	±	1	19	±	1	19	±	1	19	±	1
BMI (kg/m^2^)	23.7	±	3.9	23.6	±	3.9	23.4	±	3.5	23.3	±	3.6
Waist Circumference (cm)	81.4	±	9.9	80.6	±	10.0	80.6	±	10.8	80.2	±	9.3
Fasted Glucose (mg/dL)	87.1	±	10.7 *	85.1	±	10.0	85.7	±	11.2	85.8	±	10.2
Systolic Blood Pressure (mmHg)	118	±	11.7 *	115	±	12	116	±	13	116	±	12
Diastolic Blood Pressure (mmHg)	73	±	9	72	±	9	72	±	9	72	±	9
Fasted Triglycerides (mg/dL)	102.4	±	49.5	96.1	±	43.8	100.7	±	45.9	103.5	±	44.8 *
Fasted HDL (mg/dL)	55.5	±	15.8	54.8	±	14.1	55.6	±	14.6	56.3	±	14.4
Time spent doing Moderate Activity per week (min)	254	±	336	274	±	361	262	±	320	269	±	350
Time spent doing Vigorous Activity per week (min)	328	±	326	290	±	321	306	±	313	284	±	304
Time to Bed (hh:mm)	1:00 a.m.	±	1:01	12:11 a.m.	±	0:51	11:42 p.m.	±	0:53	11:18 p.m.	±	0:54
Time to Rise (hh:mm)	6:59 a.m.	±	1:31	7:29 a.m.	±	0:51	7:57 a.m.	±	0:53	8:48 a.m.	±	1:00
Sleep Midpoint (hh:mm)	4:01 a.m.	±	0:55	3:50 a.m.	±	0:50	3:50 a.m.	±	0:52	4:03 a.m.	±	0:54
Metabolic Syndrome Severity Score (z-score)	−0.61	±	0.02*	−0.70	±	0.02	−0.65	±	0.01	−0.63	±	0.01 *
	n		%	n		%	n		%	n		%
Gender												
Male	166		36.6%	283		31.7%	403		32.8%	359		27.8%
Female	287		63.4%	611		68.3%	827		67.2%	933		72.2%
Academic Major												
Health and/or Nutrition Major	136		30.0%	260		29.1%	301		24.5%	231		17.9%
All Other Majors	317		70.0%	634		70.9%	929		75.5%	1061		82.1%
Smoker Status												
Non-Smoker	417		92.1%	855		95.6%	1180		95.9%	1226		94.9%
Smoker	36		7.9%	39		4.4%	50		4.1%	66		5.1%
Class												
Freshmen	276		60.9%	504		56.4%	634		51.5%	729		56.4%
Sophomore	129		28.5%	259		29.0%	413		33.6%	410		31.7%
Junior	31		6.8%	79		8.8%	124		10.1%	93		7.2%
Senior	13		2.9%	45		5.0%	50		4.1%	53		4.1%
Other	4		0.9%	7		0.8%	9		0.7%	7		0.5%
Race												
Hispanic	22		5.0%	21		2.4%	41		3.4%	32		2.5%
Non-Hispanic White	381		86.8%	806		91.9%	1130		92.7%	1203		93.9%
Non-Hispanic Black/African-American	5		1.1%	9		1.0%	11		0.9%	12		0.9%
Non-Hispanic American Indian/Alaskan Native	2		0.5%	9		1.0%	2		0.2%	2		0.2%
Non-Hispanic Asian	29		6.6%	32		3.6%	35		2.9%	32		2.5%

kg, kilograms; m, meters; cm, centimeters; mg/dL, milligrams per deciliter; min, minutes; mmHg, millimeters of mercury; hh:mm, time in hours and minutes; * *p* ≤ 0.01 vs. Reference #1 continuous data are presented as adusted means±standard deviation for age, sex, semester (fall/spring), BMI, binary physical activity value, smoking status, and academic major.

**Table 3 nutrients-15-01046-t003:** Frequency of metabolic syndrome markers.

		Low Sleep (Less Than or Equal to 6.99 h)			Reference #1 (7–7.99 h)			Reference #2 (8–8.99 h)			Long Sleep (Greater Than or Equal to 9 h)		
		Mean		SD	Mean		SD	Mean		SD	Mean		SD
Metabolic Syndrome Severity Score		−0.61	±	0.02 *	−0.70	±	0.02	−0.65	±	0.01	−0.63	±	0.01 *
		n		%	n		%	n		%	n		%
Number of MetS Criteria met	0	192		42.4%	457		51.1%	610		49.6%	614		47.5%
	1	182		40.2%	305		34.1%	426		34.6%	485		37.5%
	2	59		13.0%	105		11.7%	151		12.3%	157		12.2%
	3 or more	20		4.4%	27		3.0%	43		3.5%	36		2.8%
Elevated Blood Pressure		97		21.4%	146		16.3%	225		18.3%	229		17.7%
Elevated Fasted Blood Glucose		30		6.6%	27		3.0%	46		3.7%	54		4.2%
Elevated Waist Circumference		53		11.7%	92		10.3%	133		10.8%	131		10.1%
Elevated Triglycerides		66		14.6%	94		10.5%	155		12.6%	183		14.2%
Low HDL-C		118		26.0%	239		26.7%	306		24.9%	316		24.5%

* *p* ≤ 0.01 vs. Reference #1 value; MSSS is presented as the adjusted mean ± standard deviation for age, sex, semester (fall/spring), BMI, binary physical activity value, smoking status, and academic major.

**Table 4 nutrients-15-01046-t004:** Metabolic syndrome participant demographics.

		Low Sleep (Less Than or Equal to 6.99 h)			Reference #1 (7–7.99 h)			Reference #2 (8–8.99 h)			Long Sleep (Greater Than or Equal to 9 h)		
		n = 20			n = 27			n = 42			n = 35		
Gender		n		%	n		%	n		%	n		%
	Male	10		50%	10		37%	18		43%	10		29%
	Female	10		50%	17		63%	24		57%	25		71%
		Mean		SD	Mean		SD	Mean		SD	Mean		SD
Age (years)		19	±	1	19	±	2	19	±	1	19	±	1
BMI (kg/m^2^)		30.9	±	5.7	30.6	±	6.2	28.7	±	5.6	29.2	±	5.0
Waist Circumference (cm)		100.6	±	13.0	99.8	±	14.1	95.7	±	15.6	96.6	±	11.7
Fasted Glucose (mg/dL)		91.9	±	9.1	94.5	±	22.4	89.4	±	10.1	96.3	±	16.9
Systolic Blood Pressure (mmHg)		125	±	15	124	±	14	128	±	11	123	±	14
Diastolic Blood Pressure (mmHg)		80	±	9	81	±	8	80	±	11	79	±	8
Fasted Triglycerides (mg/dL)		188.9	±	61.0	175.7	±	105.7	194.2	±	82.6	170.0	±	54.8
Fasted HDL (mg/dL)		38.3	±	11.7	40.5	±	12.8	40.0	±	12.0	44.4	±	13.6
Metabolic Syndrome Severity Score		0.56	±	0.10	0.56	±	0.08	0.51	±	0.07	0.58	±	0.08

kg, kilograms; m, meters; cm, centimeters; mg/dL, milligrams per deciliter; mmHg, millimeters of mercury; continuous data are presented as adusted means±standard deviation for age, sex, semester (fall/spring), BMI, binary physical activity value, smoking status, and academic major.

## Data Availability

Data presented in this study are available on request from the corresponding author.

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
