# Peer review of "The Relationship between Sleep Duration and Metabolic Syndrome Severity Scores in Emerging Adults"

_nutrients, 2023, doi:10.3390/nu15041046_

Round 1

Reviewer 1 Report

The main pitfall of this article is lack of a sound and strong statistical analysis. As the authors mentioned, there is a non-linear relationship between sleep and many health outcomes. Then, it is important to build an appropriate regression model controlled for confounders to show such an association. One of the most appropriate ways to build the model is the application of multivariable generalized additive model. To learn more, I advise to follow the guidelines presented here:

https://www.proquest.com/openview/e260e1515bc14ed21799c895f9b44aa5/1?pq-origsite=gscholar&cbl=18750&diss=y

Author Response

The main pitfall of this article is lack of a sound and strong statistical analysis. As the authors mentioned, there is a non-linear relationship between sleep and many health outcomes. Then, it is important to build an appropriate regression model controlled for confounders to show such an association. One of the most appropriate ways to build the model is the application of multivariable generalized additive model. To learn more, I advise to follow the guidelines presented here:

https://www.proquest.com/openview/e260e1515bc14ed21799c895f9b44aa5/1?pq-origsite=gscholar&cbl=18750&diss=y

Response: Thank you for the citation and suggestion. We have chosen to analyze the differences between sleep duration groups based on the cut-off points established in previous literature and utilized ANCOVA in order to control for the cited covariates (age, sex, BMI, physical activity level, smoking status, alcohol consumption and academic major). We have cited the reviewer’s referenced study on pg 8, line 253.

Reviewer 2 Report

Overall, an excellent and well written manuscript, exploring the role of sleep duration on metabolic syndrome severity scores in young adults. I have the following comments.

Major comments

1.     Table 3: please indicate if there were significant differences among the groups.

2.     Please consider including in the statistical methods regression analysis showing the contribution (if significant) of the other confounding factors, such as age, BMI, gender on Metabolic Syndrome Severity Scores (MSSS)

3.     The Authors very nicely compared and contrasted their results with other studies in the literature. They also did a good job in identifying the limitations of their study.  Their study would have been improved by discussing more the clinical implications of their findings.

4.     The description of the statistical analysis should be improved; crucial data is missing such as tests of normality.

5.     Another Important limitation is that, because of the open invitation for participation in the study and the lack of randomization of subjects, participants with more frequent symptoms and greater concern about sleep duration may have accepted the invitation, which would explain the relatively high prevalence of abnormal sleep duration in the sample.

Author Response

Overall, an excellent and well written manuscript, exploring the role of sleep duration on metabolic syndrome severity scores in young adults. I have the following comments.

Major comments

1. Table 3: please indicate if there were significant differences among the groups.

  1.  Thank you for the suggestion, we did run a chi-square and observed no significant differences between groups (p>0.01).

2. Please consider including in the statistical methods regression analysis showing the contribution (if significant) of the other confounding factors, such as age, BMI, gender on Metabolic Syndrome Severity Scores (MSSS)

      2.  Thank you for the suggestion. We have expanded the Results section to include the results from our regression analysis indicating the contributions of covariates, if significant (pg 7 226-227).

3. The Authors very nicely compared and contrasted their results with other studies in the literature. They also did a good job in identifying the limitations of their study.  Their study would have been improved by discussing more the clinical implications of their findings.

     3. Thank you for the suggestion; we have expanded the discussion to incorporate relevant intervention studies targeting sleep on pg.9  see 315-322.

4. The description of the statistical analysis should be improved; crucial data is missing such as tests of normality.

4. We performed Levene’s Test of equality of error variances to test for normality and observed no significant differences between all groups (p= 0.129) (pg 5, lines 185 & pg 7, lines 223-224)

  1. Another Important limitation is that, because of the open invitation for participation in the study and the lack of randomization of subjects, participants with more frequent symptoms and greater concern about sleep duration may have accepted the invitation, which would explain the relatively high prevalence of abnormal sleep duration in the sample.

5. We have expanded our discussion of the limitations to add the participation rate (>90%) from the convenience sample (pg. 9, lines 350-52). We have also expanded the description of the on-going, cross-sectional study targeting college student health (pg. 2, 85-96). Participants are informed that the study’s objective is to characterize health and lifestyle behaviors of college adults, 18-24 years of age.

Reviewer 3 Report

This is a well-written manuscript describing an original study that was aimed to examine the relationship between sleep duration and metabolic syndrome severity scores in a large group of adults in age between 18 and 24 years old.

I have only two remarks:

-authors refer to sleep duration, however, what they describe as the definition of sleep duration is time in bed (TIB) not sleep duration: “self-reported bedtime and wake time were used for calculating sleep duration”

-it should be explained more clearly why authors consider Z-scores of metabolic syndrome severity scores (MSSS) as a primary outcome and not the calculation of MSSS from table 4, which was not significant between the groups. What units have MSSS, are these points?

Author Response

This is a well-written manuscript describing an original study that was aimed to examine the relationship between sleep duration and metabolic syndrome severity scores in a large group of adults in age between 18 and 24 years old.

I have only two remarks:

-authors refer to sleep duration, however, what they describe as the definition of sleep duration is time in bed (TIB) not sleep duration: “self-reported bedtime and wake time were used for calculating sleep duration”

1. Thank you for the comment, we’ve added more to the methods to highlight this distinction (pg 4, lines 152-54). Although other cited studies have used similar questions to estimate sleep duration, we do mention this as a study limitation.

-it should be explained more clearly why authors consider Z-scores of metabolic syndrome severity scores (MSSS) as a primary outcome and not the calculation of MSSS from table 4, which was not significant between the groups. What units have MSSS, are these points?

2. The calculation of the MSSS are presented in Table 1 and used throughout the paper. As described on page 3, section 2.4, the equations provide a standardized z-score, with the mean set to zero and a range from negative infinity to positive infinity

Reviewer 4 Report

Overall: This paper examined the association between sleep duration and metabolic symptoms severity. Short sleep was associated with worse MSSS scores, as well as with specific metabolic parameters. One global comment is the use of the term “emerging adults”. Is that term common in certain areas of literature? I have never seen that term before in a publication and I wonder if “young adults” is an equivalent term. I also am concerned with the presentation of the results. Comments related to this issue and other issues can be found below.

Abstract:

1.      The age of the population seems to be a key feature of this work. However, the first sentence of the abstract doesn’t set up the purpose of the study, specifically, in young adults. That first statement is a general statement that is true for all adults.

2.      I think the term “get-up time” should be changed. In sleep research, we wouldn’t use that term. I suggest either waketime or sleep offset. To me, “get-up” is more about when you get out of bed in the morning, and not when you wake up.

Introduction:

3.      Can you spend a little more text describing reference 8? It looks like they already did this work. Based on the sentence on line 45, it looks like that previous paper focused on all adults. It would be good to further explain the gaps as it relates to that previous paper.

Methods:

4.      Is there any information on if the students were living at home or if their family was from the surrounding area? Part of the premise of this work is that this is an age when these young adults are out on their own for the first time. If they are going to school where they grew up, odds are they are living at home, or have family very close by. They are not really on their own in situations like that.

5.      I get why you had two reference categories; sleep recommendations are for 7-9 hours of sleep per night for adults. If both of those categories are referents, why not combine their numbers? It looks odd to have them separated and opens the door for questions. For example, did you analyze short and long sleep in comparison to 7-8 hours and then again in comparison to 8-9 hours? If the results were similar regardless of the reference group, then this is evidence you should combine the groups. If the results are not the same when using different referent groups, then one of them should not be a reference group and should be considered one of the “at-risk” groups.

6.      In relation to the above comment, more specific information as the referent groups should be included in the statistical analysis section.

7.      It looks like you did a subgroup analysis in those with MetS>3. Why not look at MetS as an effect modifier first? I am assuming you have a range of MSSS in those with <3 MetS. What would the results be in those without MetS?

8.      How about adjusting for year of data collection? Not only did COVID happen during data collection, but a 10-year period could lead to differences in other unmeasured societal factors that may influence the relationship.

Results:

9.      I think the tables and organization of the results needs some work. I can’t tell what tables are purely descriptive vs. adjusted values. Does table 3 show adjusted rates? If so, more work is needed to reorganize the tables and add footnotes. The same is true for Table 4. Do you have adjusted values in there somewhere? I see what appears to be some descriptive data. I strongly recommend keeping descriptive stat and adjusted values separate from each other.

Discussion:

10.  On line 208, you indicate the study brings notice of the effect of lifestyle habits on MSSS. You only focused on sleep, not other lifestyle habits. I think that (i.e., lifestyle habits) is an overgeneralization of what you actually did.

11.  Can you give a couple examples of the pathogenic pathways that you alluded to on line 216?

12.  Your references on lines 219-224 are outdated. There is no need to show data from the 80s and 90s. Your more recent evidence referenced on line 224 is still about 10 years old. Reviews on sleep and obesity with more recent evidence have been published in the last 10 years.

13.  I am not sure I understand what you mean by “Surprisingly, our study did not observe a difference between sleep duration and BMI or waist circumference”. The difference should be between two different groups, but you mention different variables.

14.  Given the findings using your two different referent groups and work you cite on line 245, I think it would be appropriate to consider 8-9 as one of the “at-risk” groups, not a referent group.

15.  I am not sure I understand your argument about sex on lines 271-273. You adjusted for sex, correct? Yes, there were more females than males. But the split was about 66 vs. 33%. There were enough males in the population. If you adjusted for sex, I don’t see how that could have compounded your results.

16.  Research indicates that, on average, adults overestimate the amount of time spent sleeping by about an hour. In other words, if adults report an average of 8 hours of sleep per night, they actually get about 7. I would highly suspect that this bias is much greater in young adults. There is a relatively high percentage of participants with >9 hours of sleep. I suspect that the misreporting in that group is disproportionally higher than in other groups. You kind of allude to this in your limitations, but I think this needs to be explicitly discussed.

Author Response

Overall: This paper examined the association between sleep duration and metabolic symptoms severity. Short sleep was associated with worse MSSS scores, as well as with specific metabolic parameters. One global comment is the use of the term “emerging adults”. Is that term common in certain areas of literature? I have never seen that term before in a publication and I wonder if “young adults” is an equivalent term. I also am concerned with the presentation of the results. Comments related to this issue and other issues can be found below.

Abstract:

  1. The age of the population seems to be a key feature of this work. However, the first sentence of the abstract doesn’t set up the purpose of the study, specifically, in young adults. That first statement is a general statement that is true for all adults.
  2. I think the term “get-up time” should be changed. In sleep research, we wouldn’t use that term. I suggest either waketime or sleep offset. To me, “get-up” is more about when you get out of bed in the morning, and not when you wake up.

Response

Abstract

  1. Thank you for the suggestions. We have edited the manuscript to highlight the increasing prevalence of MetS in adults 20-39 (pg 1, lines 36-39).
  2. Thank you. We have edited the manuscript so that “get-up time” has been changed to “wake time”. (pg1, line 8 & page 4, line 152)

    Introduction:

    1. Can you spend a little more text describing reference 8? It looks like they already did this work. Based on the sentence on line 45, it looks like that previous paper focused on all adults. It would be good to further explain the gaps as it relates to that previous paper.

Introduction:

  1. Thank you for the suggestion, we’ve added some more context to explain the importance of utilizing a narrower age range. (pg 2, lines 51-52)

Methods:

  1. Is there any information on if the students were living at home or if their family was from the surrounding area? Part of the premise of this work is that this is an age when these young adults are out on their own for the first time. If they are going to school where they grew up, odds are they are living at home, or have family very close by. They are not really on their own in situations like that.
  2. I get why you had two reference categories; sleep recommendations are for 7-9 hours of sleep per night for adults. If both of those categories are referents, why not combine their numbers? It looks odd to have them separated and opens the door for questions. For example, did you analyze short and long sleep in comparison to 7-8 hours and then again in comparison to 8-9 hours? If the results were similar regardless of the reference group, then this is evidence you should combine the groups. If the results are not the same when using different referent groups, then one of them should not be a reference group and should be considered one of the “at-risk” groups.
  3. In relation to the above comment, more specific information as the referent groups should be included in the statistical analysis section.
  4. It looks like you did a subgroup analysis in those with MetS>3. Why not look at MetS as an effect modifier first? I am assuming you have a range of MSSS in those with <3 MetS. What would the results be in those without MetS?
  5. How about adjusting for year of data collection? Not only did COVID happen during data collection, but a 10-year period could lead to differences in other unmeasured societal factors that may influence the relationship.

Methods:

  1. Thank you for the inquiry. We have edited the manuscript to include information regarding reported housing in the methods and results (pg 4, line 172-175 & pg 5, 197-198). Our participants primarily reside on campus and not with family members.
  2. This is an appreciated point. We did analyze all groups in comparison to each other through pairwise comparison, the 7-8hr group did seem to have a lower mean MSSS when compared to the 8-9hr group, implying that the 8-9hr group may be at a greater risk in this model. After consideration, we feel describing this group of young adults as “at-risk” (based upon 8-9 hrs) may be premature as previous literature has utilized 8-9 as their reference.
  3. For group comparisons, participants were categorized into 4 groups. This analysis was considered based on previous literature as described on page 4, line 154-157.
  4. We re-ran analyses on MSSS between groups with and without the participants with metabolic syndrome and significance was still achieved. As such, we chose to maintain the analyses as described.
  5. Thanks for the question, academic year was considered but did not have a significant impact on analyses so it wasn’t included as a covariate.

Results:

  1. I think the tables and organization of the results needs some work. I can’t tell what tables are purely descriptive vs. adjusted values. Does table 3 show adjusted rates? If so, more work is needed to reorganize the tables and add footnotes. The same is true for Table 4. Do you have adjusted values in there somewhere? I see what appears to be some descriptive data. I strongly recommend keeping descriptive stat and adjusted values separate from each other.

Results:

  1. Thank you for the suggestion. We have updated Table 3 and 4 to be consistent; further we have adjusted the footnote to clarify the continuous data are adjusted for confounders.

Discussion:

    1. On line 208, you indicate the study brings notice of the effect of lifestyle habits on MSSS. You only focused on sleep, not other lifestyle habits. I think that (i.e., lifestyle habits) is an overgeneralization of what you actually did.
    2. Can you give a couple examples of the pathogenic pathways that you alluded to on line 216?
    3. Your references on lines 219-224 are outdated. There is no need to show data from the 80s and 90s. Your more recent evidence referenced on line 224 is still about 10 years old. Reviews on sleep and obesity with more recent evidence have been published in the last 10 years.
    4. I am not sure I understand what you mean by “Surprisingly, our study did not observe a difference between sleep duration and BMI or waist circumference”. The difference should be between two different groups, but you mention different variables.
    5. Given the findings using your two different referent groups and work you cite on line 245, I think it would be appropriate to consider 8-9 as one of the “at-risk” groups, not a referent group.
    6. I am not sure I understand your argument about sex on lines 271-273. You adjusted for sex, correct? Yes, there were more females than males. But the split was about 66 vs. 33%. There were enough males in the population. If you adjusted for sex, I don’t see how that could have compounded your results.
    7. Research indicates that, on average, adults overestimate the amount of time spent sleeping by about an hour. In other words, if adults report an average of 8 hours of sleep per night, they actually get about 7. I would highly suspect that this bias is much greater in young adults. There is a relatively high percentage of participants with >9 hours of sleep. I suspect that the misreporting in that group is disproportionally higher than in other groups. You kind of allude to this in your limitations, but I think this needs to be explicitly discussed.

Discussion:

  1. Thank you for the suggestion; we have edited the manuscript on pg 8, line 246, to be more specific to the focus on sleep.
  2. Thank you for the suggestion, we have expanded our discussion on the pathogenic pathways on pg 8, line 254-6.
  3. Thank you for the comment. We were attempting to be complete in our analysis of the literature. We’ve updated the paragraph to focus on more recent evidence (pg 8, line 258).
  4. Thank you for noting this. We’ve edited the manuscript to correct this on page 8, lines 285-6, “Surprisingly, our study did not observe significant difference in BMI or waist circumference between sleep duration groups.”
  5. Thank you for the suggestion. We left this as a reference group because it is still within the sleep recommendations for young adults.
  6. Thank you for the suggestion, the manuscript has been edited and comment removed.
  7. Thanks for the comment. We have expanded the limitation section to explicitly include the possibility of overestimation and referenced previous work in middle-aged adults (pg 10, line 367)

Round 2

Reviewer 4 Report

The authors addressed most of my comments or justified why changes weren't needed for certain things. However, I still have a few remaining minor items:

1.      For my comment one from the abstract, I appreciate the change you made to the introduction as it is helpful. However, I was thinking of a needed change to the abstract. Maybe you could add something along the lines of “even in young adults” to the end of the first sentence of the Abstract?

2.      I noticed that my question in my overall comment wasn’t addressed. Please see that previous comment about use of the term “emerging adults”. Is that an accurate term or would “young adults” be better?

3.      In the next text on line 62 within the introduction, what does “low sleep” mean?

4.      Thank you for updating the data. I suggest putting the actual covariates in the footnotes as opposed to just saying “adjusted”. Tables, much like the abstract, should be standalone documents. Readers should not have to refer to the manuscript text to make sense or confirm something in the tables. The tables should contain everything needed to make a clear picture of the results being presented in that table.

Author Response

  1. Thank you for the suggestion, we’ve adjusted the manuscript accordingly. Please see page 1, line 11.
  2. Thank you for the concern, the term “Emerging adults” has been used in previous literature to accurately describe this population. In the aim to stay consistent with that literature we chose to use this term rather than “young adults”, we have updated the manuscript with citations (pg 2 line 53).

  3. In the context of this citation, “Low Sleep” refers to 4 hours of sleep per night, we have edited the manuscript to clarify this point (pg 2, line 62).

  4. Excellent point, we have adjusted the tables to include the actual covariates in the footnotes as well.